# Recovery Behavior of the Macro-Cracks in Elevated Temperature-Damaged Concrete after Post-Fire Curing

**DOI:** 10.3390/ma15165673

**Published:** 2022-08-18

**Authors:** Lang Li, Yao Chen, Chao He, Chong Wang, Hong Zhang, Qingyuan Wang, Yongjie Liu, Guomin Zhang

**Affiliations:** 1Failure Mechanics & Engineering Disaster Prevention, Key Laboratory of Sichuan Province, College of Architecture & Environment, Sichuan University, Chengdu 610065, China; 2Key Laboratory of Deep Underground Science and Engineering, Ministry of Education, Sichuan University, Chengdu 610065, China; 3Civil and Infrastructure Engineering Discipline, School of Engineering, RMIT University, Melbourne, VIC 3001, Australia

**Keywords:** crack, concrete, recovery, elevated temperature, post-fire curing

## Abstract

Studying the recovery of fire-damaged concrete is of huge economic and environmental significance. The recovery of thermal-induced cracks of fire-damaged concrete leads to the recovery of strength after post-fire curing. To identify the crack recovery behavior of fire-damaged concrete after post-fire curing and its relationship with the recovery of strength, in this study, concrete samples exposed to 400, 600, and 800 °C were treated with the post-fire curing process. The compressive strength recovery was investigated, as well as the crack recovery in terms of the crack length. Moreover, the recovery of the cracks was studied and divided into the categories of mortar cracks and mortar-aggregate interfacial cracks. The results indicate that, after being exposed to high temperatures, the interfacial crack was the main type of crack, and it could clearly be recovered by post-fire curing. The recovery of compressive strength mainly resulted from the recovery of interfacial cracks. The findings of this study can provide practical guidance for the application of post-fire curing to the recovery of fire-damaged concrete structures.

## 1. Introduction

Although invented hundreds of years ago, concrete is still one of the most widely used construction materials in the world because of its excellent comprehensive performance, including its lower cost, ease to manufacture, excellent durability and mechanical properties, incombustibility, and good fire resistance. During its life of service, concrete encounters high temperatures occasionally, for instance in fire disaster. Even with good fire resistance, concrete also deteriorates in terms of its service abilities, such as a loss of strength, decrease in the elastic modulus, degradation of durability, and so on [1,2]. After fire exposure, constructions are supposed to be evaluated and repaired as soon as possible. The most convenient way to do this is to remove the damaged layers and cast the structure with fresh concrete. Recently, thanks to the development of fiber-reinforced plastic (FRP) [3,4,5], the use of FRP to repair fire-damaged concrete was also reported [6].

It is believed that the mechanical and physical changes during exposure to high temperatures is the cause of the deterioration of the service abilities of concrete [7,8,9]. These changes include thermal-induced cracks and pores [10], the evaporation of water, and the decomposition of hydrated cement [8]. The mechanical changes and chemical changes are correlative. For example, the water evaporates at temperatures over 100 °C, and with the increasing temperature, the pressure of the water vapor increases. This can cause micro-cracks around the pores or even spalling [10,11], and, on the contrary, it also creates an internal autoclaving condition, resulting in the further hydration of non-hydrated cement up to 300 °C [12]. At high temperatures, the hydrated cement decomposes, which causes the shrinking of the cement matrix, while the aggregate expands, and the discrepancy of the thermal deformation introduces cracks in the cement matrix and the matrix-aggregate interfaces.

During exposure to high temperatures, the chemical changes in the cement matrix are complicated. The calcium silicate hydrate (C-S-H) gel starts dehydrating to shorter chains at 200 °C [13,14] and completely dehydrates to calcium silicate (C_n_S) at 750 °C [15]. Additionally, up to 300 °C, the non-hydrated cement grains can further hydrate to C-S-H and produce calcium hydroxide (CH) as the by-product because of the internal autoclaving condition [16]. The CH dehydrates to calcium oxide (CaO) around 500 °C, and the calcium carbonate (CaCO_3_) decomposes to CaO at over 750 °C [17]. The chemical changes that occur at high temperatures mainly refer to the decomposition of the cement matrix, which results in the destruction of the cementitious system of the concrete. However, it also provides the concrete with the ability to recovery after high temperature exposure.

In the 1970s, Crook and Murray [18] first reported that fire-exposed concrete can regain its strength by immersion in water after fire exposure. They also found that the large thermal-induced pores were segmented to create smaller ones after the immersion. With a scanning electron microscope, Lin et al. [19] found that some new hydrated products were formed in the fire-damaged concrete after water curing. Poon et al. [20] reported that not only the strength, but also the permeability, can also be recovered, and they further proved that the thermal-induced cracks can be healed by water curing after fire exposure, and this healing process was named post-fire curing or recuring. Poon et al. [20] also found that the pozzolanic materials, such as fly ash, silica fume, and metakaolin, are beneficial for the recovery of strength after post-fire curing. They also believed that, for concrete exposed to temperatures lower than 600 °C, post-fire curing is an effective and economic recovery method.

The recovery of fire-damaged concrete after post-fire curing is the result of rehydration of the dehydrated products [16]. The rehydration products can fill the thermal-induced cracks and pores to reconstruct the cementitious network, and then the properties of the concrete can be recovered. During the post-fire curing process, the dehydrated C_n_S, mainly C_2_S, can be rehydrated to C-S-H [15], and the CaO can be rehydrated to CH [21]. Moreover, some other products, such as ettringite, are also formed during the process [15,22]. Generally, the hydration of CaO to CH is considered harmful to the microstructure of concrete since its volume expansion causes further damage. However, for fire-damaged concrete, the thermal-induced cracks and pores provide space for the formation of CH and, consequently, the damage effect is eliminated. Furthermore, for concrete formed with pozzolanic materials, the CH can react with these materials and aid the recovery [22]. It has been observed that the rehydration products can fill up thermal-induced cracks and pores, consequently healing the microstructure and recovering the properties of the concrete [19,20]. It has been reported that, after post-fire curing, the healing of cracks can also be observed at the macro level. Li et al. [22] reported that the macro-crack was shorter in total length after post-fire curing, and that the decrease in the crack length is positively related to the strength recovery. Generally, several methods can be used to detect the cracks in the concrete. Henry et al. [23] has reported an epoxy injection method to detect the cracks inside the concrete. Micro-X-ray-computed (μCT) tomography has also been reported to detect the cracks in concrete [24,25]. Compared with other methods, μCT is an excellent method for detecting cracks in concrete, since it can detect micrometer-scale cracks inside the concrete without causing damage to the samples, and it can reconstruct the 3D structure of the cracks [16]. In contrast, in this paper, we focused on the macro-cracks. Therefore, a method based on image analysis was adopted, as we have reported previously [22].

Our previous study [26] indicated that the recovery by post-fire curing resulted from the recovery of the concrete’s mortar-aggregate interfacial strength. However, the relationship between the strength recovery and the crack recovery is still unclear. Therefore, in this study, the recovery of cracks after post-fire curing was investigated. The relationship between the recovery of the cracks, especially the interfacial cracks, and the recovery of strength was discussed. The results of this study can help to improve our understanding of the mechanism behind the recovery of fire-damaged concrete after post-fire curing.

## 2. Materials and Methods

### 2.1. Concrete Mixture and Curing

In this study, 42.5R ordinary Portland cement from the brand E’sheng, which was blended with low-calcium fly ash, was adopted as the cementitious materials, and the weight ratio of the cement to fly ash was 4:1. The type of cement and fly ash used in this study was the same as we used in our previous study [27], and the chemical composition, determined with energy disperse spectroscopy (EDS), is shown in Table 1. Local river sand around Chengdu was used as the fine aggregate. The coarse aggregates were acquired from local gravel in the size range of 4.75 to 20 mm, and the grain size distribution of the coarse aggregate is shown in Figure 1. To reduce the water consumption, a powdery polycarboxylic superplasticizer (S.P) was added with a dosage of 0.3 wt% of the cementitious materials. The water to cementitious material ratio was 0.29. The mixture proportions of the fresh concrete are shown in Table 2.

The raw materials were blended in a laboratory blender, and then the fresh concrete was cast in cubic molds with edges of 100 mm with two equal layers, and each layer was vibrated on the vibration table for 30 s. After smoothing the surface of each specimen with a trowel, the specimens in the molds were covered with a layer of thin plastic film to keep the moisture from evaporating. The specimens were demolded within 24 h after casting, then cured in a chamber at 20 °C with a relative humidity (RH) of 95% for 28 days. Then, the specimens were cured in the laboratory environment for 90 days before the high-temperature exposure. The 28-day compressive strength of the specimens was 74.6 MPa, and the 90-day compressive strength was 79.0 MPa.

### 2.2. High-Temperature Exposure and Post-Fire Curing

After curing for at least 90 days, the specimens were subjected to high-temperature exposure. The high-temperature exposure was conducted with an electric furnace that could control the temperature precisely using pre-set programs. The specimens were heated at the rate of 5 °C/min to 400 °C, 600 °C, or 800 °C. After reaching the target temperature and being maintained at the target temperature for 30 min, the specimens were cooled down to room temperature in the furnace.

To acquire the temperature history of the specimens during high-temperature exposure, a type-K thermocouple with a diameter of 4 mm was placed in the middle of one specimen which was exposed to each temperature. The thermocouples were added to the specimens after 28 days of curing. A hole with a diameter of 6 mm, which was 50 mm deep, was drilled into the specimen from the middle of one of the surfaces. After adding the thermocouples, the holes were filled with the same mortar and continued curing until the time of the high-temperature exposure. During the high-temperature exposure, the thermocouples were linked to the computer through a data exchange device, which enabled us to acquire the temperature data every 7 s. The development of the temperature of the furnaces and the specimens is shown in Figure 2.

The post-fire curing process was carried out according to the authors’ reported studies [1,2]. After cooling down to room temperature, the specimens were submerged in water for 24 h and then cured in a chamber at 20 °C with the RH of 95% for another 29 days, before the following tests were conducted. Therefore, the whole post-fire curing lasted for 30 days. After the post-fire curing, the specimens were immediately subjected to the following tests.

### 2.3. Thermal Expansion

It is believed that the unequal coefficient of the thermal expansion between the aggregate and the cement matrix is responsible for the thermal-induced cracks in concrete subjected to high temperatures. Therefore, in this study, a device was designed to determine the thermal expansion of the concrete and the mortar, following the concept of the previous study [28], as shown in Figure 3. The specimen used for the thermal expansion test was a 40 mm × 40 mm × 160 mm cuboid. The concrete specimens were manufactured according to the mixture shown in Table 2, and the mortar specimens were also manufactured according to Table 2, without the coarse aggregate, the ratio of the other raw materials was kept the same as that of the concrete.

The device consisted of a gypsum foundation, two ceramic rollers, two ceramic rods, and two micrometer gauges. When testing the thermal expansion of the specimen, the gypsum foundation, ceramic rollers, and specimens were placed in the furnace, the micrometer gauges were immobilized outside the furnace, and the ceramic rods were inserted into the furnace with the other ends connected to the micrometer gauges, as shown in Figure 3.

The testing specimen was placed on the two ceramic rollers to ensure that the specimen could expand freely at high temperatures. One end of the specimen was in contact with the gypsum foundation, while the other end was connected to the micrometer gauge by the ceramic rod. Then, the thermal expansion of the specimen and the connected ceramic rod were measured with the micrometer gauge. Another ceramic rod, with one end in contact with the gypsum foundation and the other end connected to the micrometer gauge, was also placed on the setup in order to measure the expansion of the ceramic rod. As shown in Figure 3, the effect tested with micrometer gauge 1 was the expansion of the specimen and ceramic rod 1, while that tested with micrometer gauge 2 was the expansion of ceramic rod 2. Therefore, the thermal expansion of the specimen at temperature *T* can be calculated as follows:*TE_T_* = (*D*1*_T_* − *D*2_20_) − (*D*2*_T_* − *D*2_20_) · *l*_1/_*l*_2_(1)
where *TE_T_* is the thermal expansion of the specimen at temperature *T*, *D*1*_T_* is the result of micrometer gauge 1 at temperature *T*, *D*1_20_ is the result of micrometer gauge 1 at 20 °C, *D*2*_T_* is the result of micrometer gauge 1 at temperature *T*, *D*2_20_ is the result of micrometer gauge 2 at 20 °C, and *l*_1_ and *l*_2_ are the lengths of the ceramic rods 1 and 2 in the furnace, respectively.

During the thermal expansion testing, the furnace was set to heat at 10 °C/min to 200 °C, 400 °C, 600 °C, and 800 °C, and to maintain the target temperatures for 50 min in order to ensure a stable temperature field inside the specimen. For both mortar and concrete, 3 specimens were tested and the mean value of three specimens was adopted as the thermal expansion at the testing temperatures.

### 2.4. Compressive Strength

The compressive strength of the specimens at 28 days and 90 days, after the high-temperature exposure and after post-fire curing was tested. The compressive strength of the specimens was tested with an electrohydraulic servo-controlled compression testing machine with the loading capacity of 2000 kN at Sichuan University, China. The loading speed was set as 1 kN/s.

### 2.5. Crack Detection and Analysis

The macro-cracks inside the concrete were identified by analyzing the images of the cross-sections of the specimens. To acquire the images of the cross-sections, the 28-day-cured specimens were cut with a concrete cutting machine, as shown in Figure 4. During cutting, the blade of the machine was cooled with flowing water to avoid further damage to the specimens. Then, the two parts of a specimen were recombined using the same mortar along the edge of the curing surfaces, and the curing continued until the high-temperature exposure was achieved. After the exposure and post-fire curing, the images of the cross-sections were taken with a digital camara. Since the images for the crack detection and analysis were taken using the same cross-section before and after the post-fire curing, the results were more comparable than those from our previous study [22].

The cracks were detected by hand using the pencil tool in the Adobe Photoshop program. Although hand detection is low in efficiency, it is still more accurate than the program in detecting the cracks in the cross-sections, in which a lot of aggregates exist. After opening the detecting image on Adobe Photoshop, a new layer was built to copy the cracks using the pencil tool by hand-drawing. The cracks at the interfaces and in the mortar were drawn in different colors for their distinction. The interface cracks were drawn in red (in RGB mode, this was absolute red, and the value of each channel was R = 255, G = 0, B = 0), while the mortars were in blue (R = 0, G = 0, B = 255), as shown in Figure 5a.

## 3. Results

### 3.1. Compressive Strength

As shown in Figure 6, after the high temperature exposure, the compressive strength obviously decreased, and the decrease was greater with the increasing exposed temperature. The compressive strength decreased from 79 MPa, when not exposed, to 64.5, 48.5, and 27.7 MPa after being exposed to 400, 600, and 800 °C, respectively. After post-fire curing, the compressive strength of the specimens exposed to 600 and 800 °C was significantly recovered, changing from 48.5 and 27.7 MPa to 58.1 and 35.7 MPa, respectively. Although the residual compressive strength of the specimens exposed to 400 °C was the highest compared to those exposed to 600 and 800 °C, its recovery after the post-fire curing was also not obvious. As we reported previously, the amount of dehydrated products was not enough to foster any effective recovery of the compressive strength after post-fire curing from 400 °C [26]. The temperature of 400 °C was not high enough to ensure the dehydration of CH to create CaO [14], and there was not enough dehydrated C_n_S for the rehydration during the post-fire curing process [21].

### 3.2. Thermal Expansion

With the method described in Section 2.3, the thermal strain of the concrete and mortar specimens was measured. With the thermal expansion taken into account, the thermal strain of the specimens can be calculated as follows:(2)$$\varepsilon_{T}=\frac{TE_{T}}{l_{0}}$$
where *ε_T_* is the thermal strain at temperature *T*, *TE_T_* is the thermal expansion of the specimen at temperature *T*, and *l*_0_ is the length of the specimen at an ambient temperature, which was 160 mm in this study.

The thermal strain of both the concrete and mortar is shown in Figure 7a, and the results were also compared with those of the previous study [29], as shown in Figure 7b. It is obvious that the experimental results match the results reported by previous studies closely, which indicates that the experimental method adopted in this study is reliable.

As shown in Figure 7a, for both the concrete and mortar, the thermal strain increased with the temperature. The thermal strain of the concrete and mortar was almost the same at 200 °C, but that of concrete was obviously higher than the mortar after 200 °C, which correlates with the results reported by Schneider [29] in 1988. At a temperature higher than 200 °C, because of the higher coefficient of the thermal expansion (CTE) of the concrete [30], the thermal strain of the concrete increased more with the increasing temperature than the mortar. From Figure 7a, it is easily observed that the thermal strain of the concrete increased with a similar rate at 400 °C as at 200 °C, and it kept increasing at 600 °C and 800 °C with a higher rate. Moreover, the increase rate at 600 °C and 800 °C seemed to be unchanged. Comparatively, for the mortar, the thermal strain increased differently from that of the concrete. With a similar increase rate to that of the concrete at 200 °C, the increase rate of mortar obviously dropped at 400 °C, rose at 600 °C, and reached the same level as that of the concrete at 800 °C.

### 3.3. Recovery of the Cracks

The crack lengths in the cross-sections of the specimens after the high-temperature exposure and post-fire curing are shown in Figure 7. It is obvious that the length of the cracks increased with the increasing exposed temperature. With the same temperature increase, the difference in the crack lengths between 600 °C and 400 °C was much greater than that between 800 °C and 600 °C. After being exposed to 400 °C, the crack length was 436 mm, and it increased to 1229 and 1582 mm after being exposed to 600 °C and 800 °C. As shown in Figure 8, the crack was significantly recovered after the post-fire curing. The crack length recovered from 436 mm to 306 mm after post-fire curing after exposure to 400 °C, and from 1229 mm to 590 mm after 600 °C, from 1582 mm to 1148 mm after 800 °C. In terms of the crack length after post-fire curing, the specimens exposed to 400 °C showed the best recovery, since the crack length after high-temperature exposure was the lowest. In terms of the decrease in the crack length, those post-fire samples cured after exposure to 600 °C showed the best recovery, with an average crack length decrease of 639 mm, as shown in Figure 8a. When compared with the recovery of the compressive strength, shown in Figure 6, we can observe that the recovery of the crack was positive related to the recovery of the compressive strength. It is easy to understand this relationship, since both the recovery of the crack and recovery of the strength were consequences of rehydration during the post-fire curing process. As discussed in our previous study [26], the recovery of concrete after post-fire curing is positively related to the rehydration, while negatively related to thermal-induced damage. Higher exposed temperatures resulted in severer damage but more products available for rehydration during post-fire curing. These two factors appeared to reach a balance at 600 °C, the temperature at which the best recovery was observed.

To discover the behavioral differences between the interfacial and mortar cracks, the lengths of the two different types of cracks were calculated, as described in Section 3. As shown in Figure 8b, the length of the interfacial crack was 400 mm after being exposed to 400 °C, and the length increased to 1068 mm and 1116 mm for those cracks exposed to 600 °C and 800 °C, respectively. For the specimens exposed to 400 °C, the length of the mortar crack was 36 mm, and the lengths were 161 mm and 466 mm for those exposed to 600 °C and 800 °C. Although the lengths of both the interfacial and mortar cracks increased with the increasing exposed temperature, the greatest increase in the two types of cracks occurred at different temperatures. For the interfacial cracks, from 400 °C to 600 °C, the length increased from 400 mm to 1068 mm with a 668 mm increment, and just a 48 mm increment from 600 °C to 800 °C. Comparatively, for the mortar crack, the largest increment occurred from 600 °C to 800 °C, increasing from 161 mm to 466 mm.

It is obvious that, among all the specimens exposed and post-fire cured at all the temperatures, most of the cracks were interfacial crack, which takes 70% to over 90% of the crack in all the specimens. Moreover, the proportion of interfacial cracks decreased with the increasing exposed temperature. For the specimens exposed to 400 °C, the length interfacial cracks comprised 91.7% of the cracks, and this dropped to 86.9% and 70.5% for those cracks exposed to 600 °C and 800 °C, respectively. In other words, the proportion of the mortar cracks increased with the increasing temperature. This indicates that, with the increase in the exposed temperature, the damage to the mortar became more severe because of the thermal disparity between the aggregate and the mortar, which caused mortar cracks [30,31,32].

As shown in Figure 7a, at a temperature higher than 200 °C, the thermal strains of mortar and concrete were quite different. At a temperature higher than 200 °C, the thermal expansion of the concrete was higher than that of the mortar. Consequently, cracks were induced in the mortar during high-temperature exposure and remained after cooling down. At a higher temperature, the differences in the thermal strain between the concrete and mortar became larger, as shown in Figure 7a. The expansion of the aggregate was constrained by the mortar around it, as shown in Figure 9a; consequently, tensile stress was induced in the mortar, as shown in Figure 9b. Meanwhile, compressive stress was induced in the aggregate. With the temperature rose, the tensile stress increased, and cracks were induced in the mortar, and the compressive stress induced cracks in the aggregate, as shown in Figure 9c. After cooling down, the aggregate shrunk while, because of the cracks in the mortar, the mortar around the aggregate was not able to shrink with the aggregate; thus, cracks were induced at the interface of aggregate and mortar, which are named interfacial cracks.

As mentioned above, the recovery of concrete after post-fire curing is the result of rehydration of dehydrate products that are produced during high-temperature exposure. At high temperatures, the hydrated products of the cement matrix are dehydrated to form CnS, CaO, etc., which means that the dehydrated products can only be found in the cement matrix. As we know, there are two sides to a crack. For the mortar crack, both sides are mortar, while for the interfacial crack, only one is. Thus, better recovery can be observed in the mortar cracks.

However, the relationship between the recovery of the crack and the recovery of the compressive strength indicates that the recovery of interfacial cracks is more related to the recovery of the strength at 400 °C and 600 °C, as shown in Figure 10. In our previous study [26], it was found that the recovery of strength after post-fire-curing mainly resulted from the recovery of mortar-aggregate interfacial cracks. With the results from the previous study and this study taken together, it can be concluded that the recovery of the strength mainly results from the recovery of the interfacial strength, which is specific to the recovery of an interfacial crack. Moreover, at 800 °C, the recovery of mortar cracks is obvious, which also benefits the recovery of the concrete strength.

## 4. Conclusions

In this study, the recovery of thermal-induced cracks in concrete after post-fire curing was studied, the thermal expansion of the concrete and mortar was measured, and the strength recovery was also investigated. Based on the experimental results, the main conclusions can be drawn as follows:
After being exposed to high temperatures between 400 °C and 800 °C, the compressive strength of concrete obviously decreased, and the decrease grew with the increase in the exposed temperature. And after post-fire curing, the compressive strength of concrete can be recovered, but it cannot recover to the level before the high-temperature exposure. The recovery was not obvious for those samples exposed to 400 °C, but it was significant for those exposed to 600 °C and 800 °C. The strength after recovery decreased with the exposed temperature increase.After being exposed to a temperature over 400 °C, macro-cracks were found inside the concrete, and the crack length grew with the exposed temperature increase. After post-fire curing, both mortar cracks and interfacial cracks can be recovered, while the recovery of interfacial cracks is much more attainable. For the samples exposed to 400 and 600 °C, the recovery of the interfacial cracks benefitted more from the strength recovery, since the relationship between the crack recovery and the strength recovery indicates that the interfacial crack recovery is related to the recovery of strength.In terms of the strength recovery and macro-crack recovery, concrete subject to temperatures lower than 600 °C can be significantly recovered after-post-fire curing but cannot recovered to the original level. We suggest using post-fire curing as a subsidiary method in repairing fire-damaged concrete.

## Figures and Tables

**Figure 1 materials-15-05673-f001:**
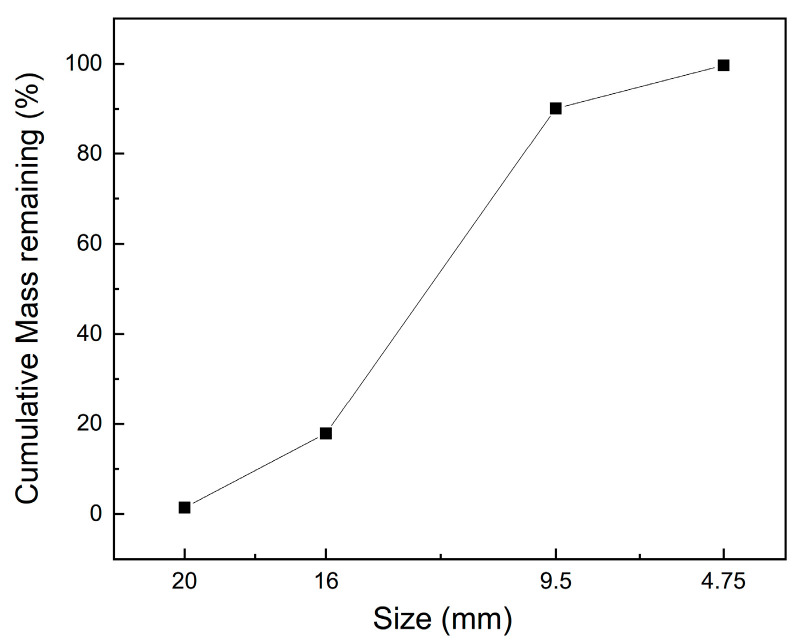
Grain size distribution of the coarse aggregate.

**Figure 2 materials-15-05673-f002:**
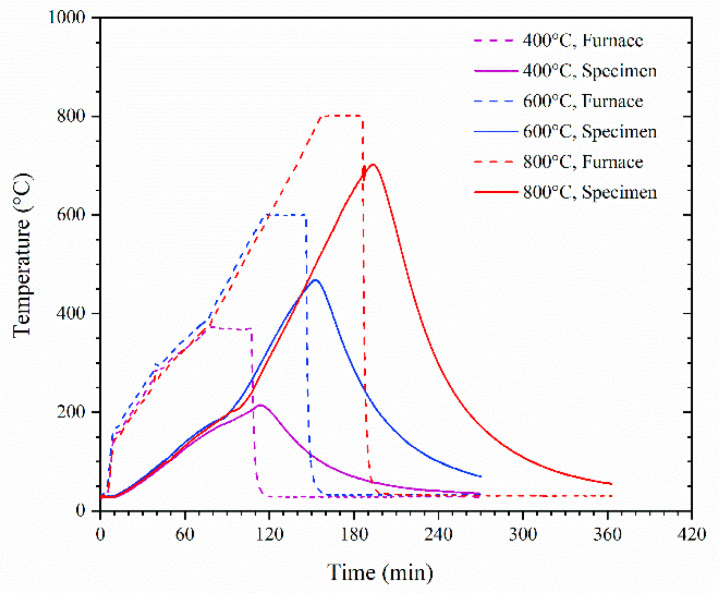
Development of the temperature of the specimens exposed to different high temperatures.

**Figure 3 materials-15-05673-f003:**
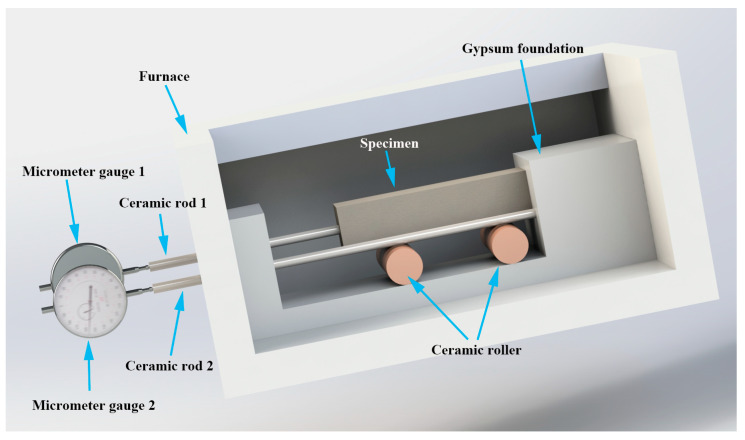
Device used to determine the thermal expansion of the concrete and mortar.

**Figure 4 materials-15-05673-f004:**
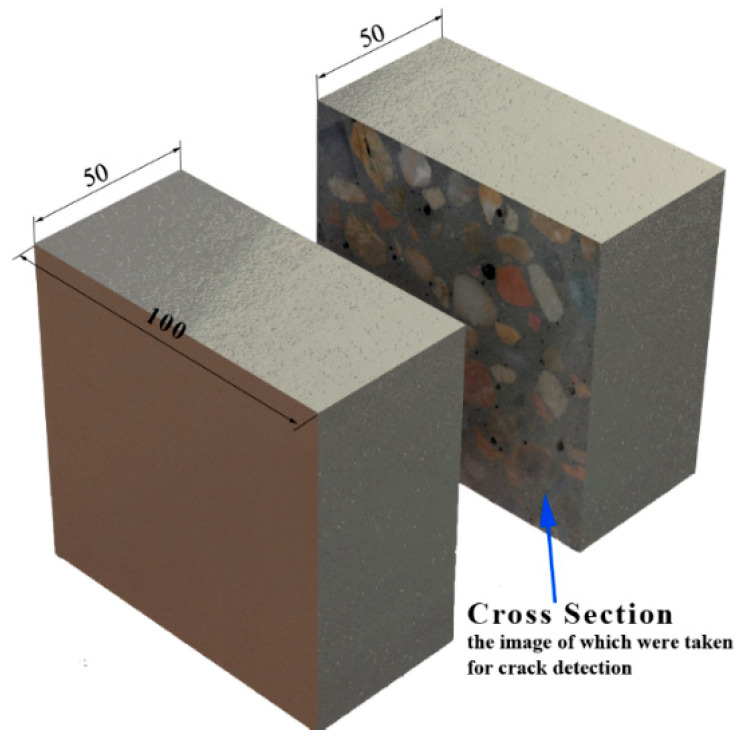
Cutting of the concrete specimen.

**Figure 5 materials-15-05673-f005:**
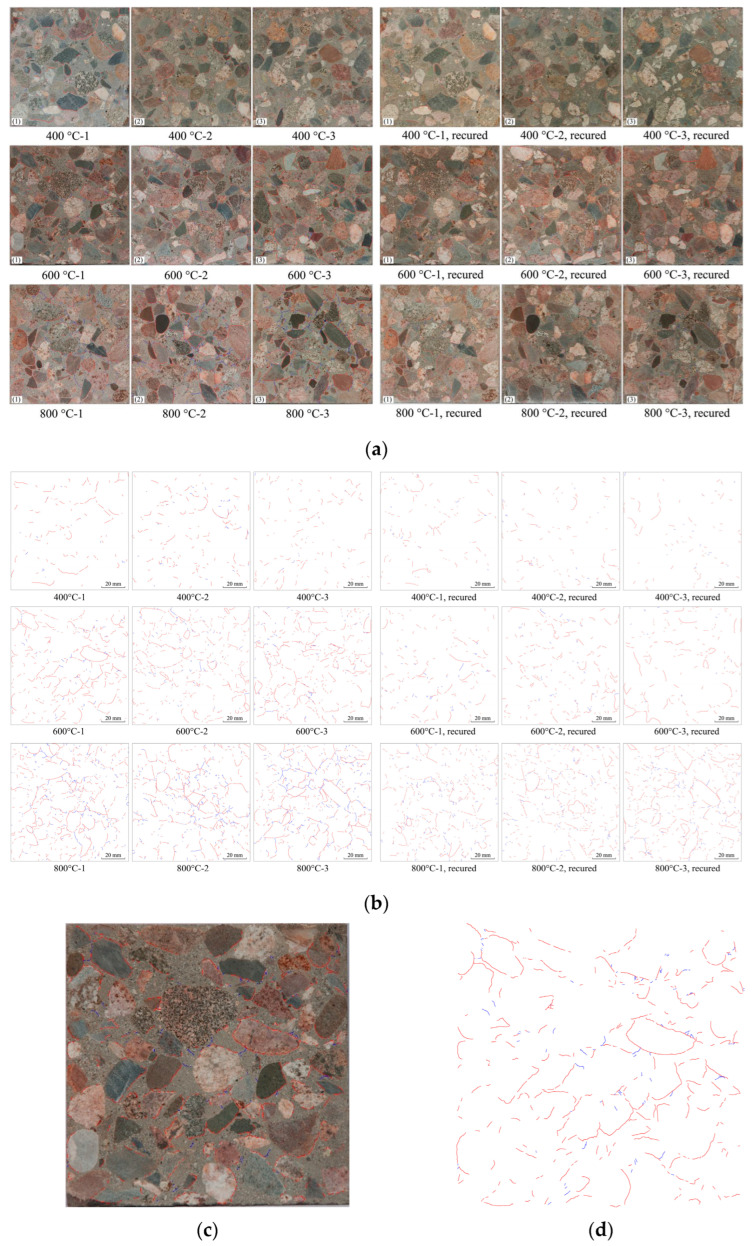
Drawn cracks in the images of the specimens’ cross-sections, with the red lines representing the interface cracks and blue lines representing mortar cracks: (**a**) cracks drawn within the raw pictures; (**b**) cracks recognized with the user-defined MATLAB function; (**c**) example of a crack drawing; and (**d**) example of cracks recognized with the user-defined MATLAB function.

**Figure 6 materials-15-05673-f006:**
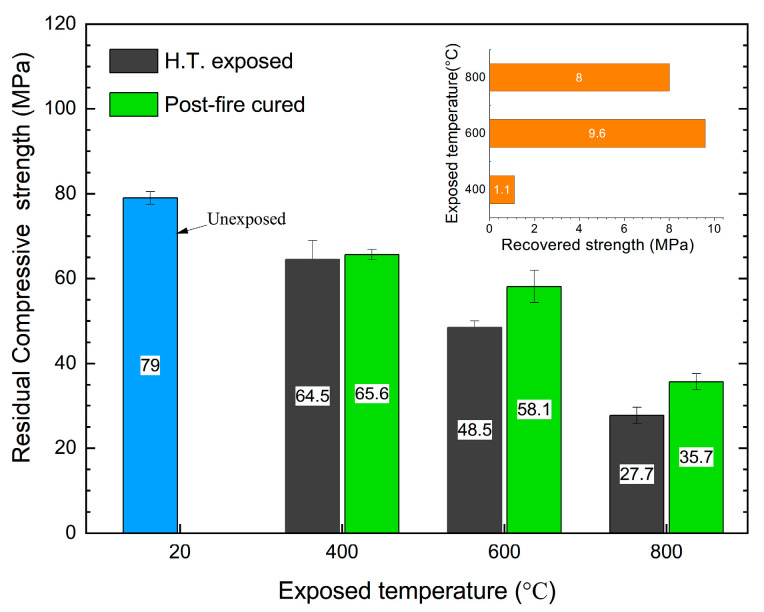
Residual compressive strength of the specimens after high-temperature exposure and post-fire curing.

**Figure 7 materials-15-05673-f007:**
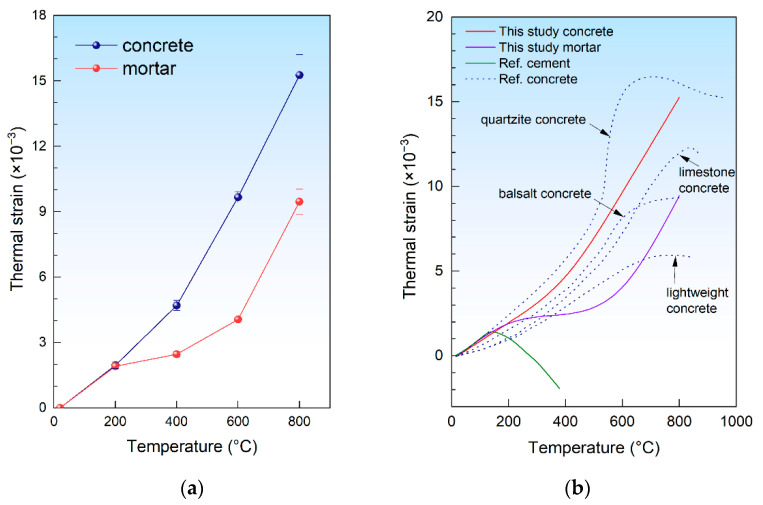
Thermal strain of the concrete and mortar: (**a**) experimental results of the thermal strain measured in this study; (**b**) thermal strain compared with the reference [29].

**Figure 8 materials-15-05673-f008:**
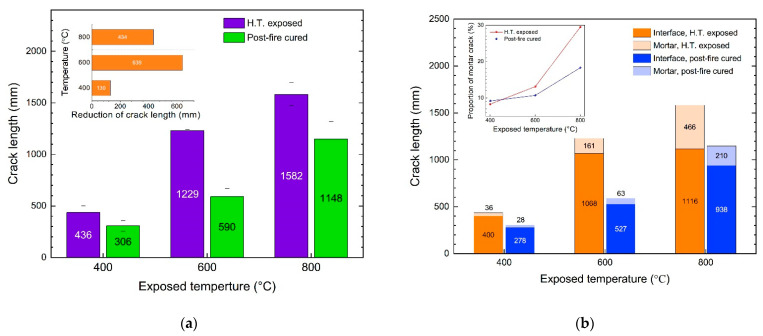
Recovery of the cracks in the cross-sections after post-fire curing: (**a**) recovery of the total crack length; and (**b**) recovery of different types of cracks.

**Figure 9 materials-15-05673-f009:**
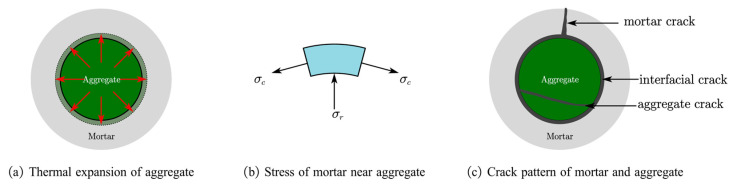
Schematic diagram of the cracks induced by the nonuniform thermal expansion between the aggregate and mortar.

**Figure 10 materials-15-05673-f010:**
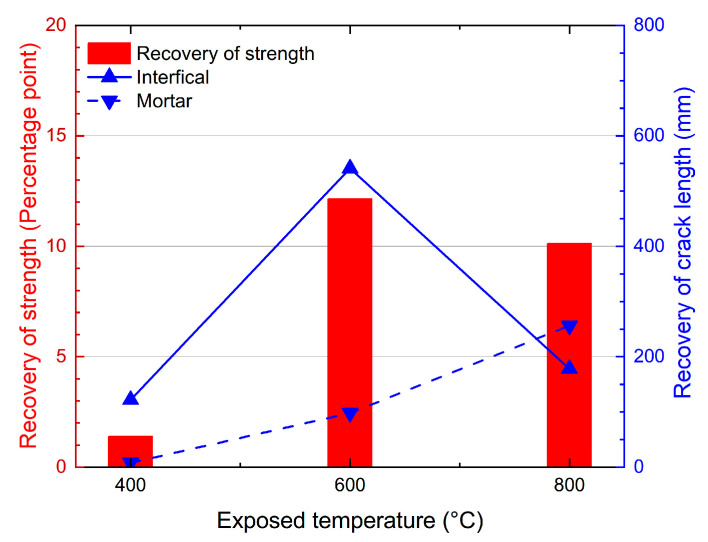
Relationship between the recovery of the crack and the recovery of the compressive strength.

**Table 1 materials-15-05673-t001:** Chemical composition of the cement and the fly ash (percentage in mass).

Chemical Composition	SiO_2_	Al_2_O_3_	CaO	Fe_2_O_3_	MgO	K_2_O	SO_3_	TiO_2_	Na_2_O	LOI ^a^
Cement	17.78	2.49	63.67	2.5	3.09	0.46	4.77	0.80	0	4.53
Fly ash	49.05	26.40	5.20	4.64	3.72	4.85	2.00	1.16	0.80	2.83

^a^ Loss on ignition.

**Table 2 materials-15-05673-t002:** Mixture proportions of the fresh concrete (kg/m^3^).

Cement	Fly Ash	Fine Aggregate	Coarse Aggregate	S.P.	Water
412	103	571	1162	1.545	149

## Data Availability

Not applicable.

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
