# Peer review of "Recovery Behavior of the Macro-Cracks in Elevated Temperature-Damaged Concrete after Post-Fire Curing"

_materials, 2022, doi:10.3390/ma15165673_

Round 1
Reviewer 1 Report
The authors have to clarify more the reason for this statement "After exposed to the temperature over 400 °C, macro cracks were found inside the concrete, and the crack length grows with the exposed temperature. The mismatch of thermal expansion between the mortar and aggregate causes the cracks. The mortar-aggregate interfacial crack is the main type of crack." at the conclusion otherwise, the conclusion will sound useless
Authors could relate that with the coefficient Of expansion of aggregate or the resistance of an aggregate to elevated temperature or post firing (testing the ingredients of concrete and mortar before mixing or if there is any existing data to relate to their conclusion.
Author Response
Thanks for your suggestion. After carefully and deep consideration, we agree with you that this point of conclusion is not strong enough, and the similar conclusion has been reported by former articles (Fu et al, 2004 in CCC, Reference 31 and 32 in the revised manuscript). After comprehensively considering the comments from other reviewers, we have finally integrated this point of conclusion with the third one. The revised conclusions are quoted below:
In this study, the recovery of the thermal-induced cracks in concrete after post-fire-curing was studied, the thermal expansion of the concrete and mortar was measured, the strength recovery is also investigated. Based on the experimental results, the main conclusions can be drawn as follows:
- After exposed to high temperatures between 400 °C and 800 °C, the compressive strength of concrete obviously decreased, and the decrease grows with the exposed temperature. And after post-fire-curing, the compressive strength can be recovered, but cannot recover to the level before the high-temperature exposure. The recovery is not obvious for those exposed to 400 °C, but significant for those exposed to 600 °C and 800 °C. The strength after recovery decreases with the exposed temperature.
- After exposed to the temperature over 400 °C, macro cracks were found inside the concrete, and the crack length grows with the exposed temperature. After post-fire-curing, both the mortar crack and interfacial cracks can be recovered, while the recovery of interfacial crack is much more obvious. For 400 and 600 °C, the recovery of interfacial crack benefits more for the strength recovery since the relationship between crack recovery and the strength recovery indicates that the interfacial crack recovery is more related to the recovery of strength.
- In the aspect of the strength recovery and macro crack recovery, the concrete subject to the temperature lower than 600 °C can be significant recovered after-post-fire-curing but cannot recovered to the original level. It is suggested to use the post-fire-curing as a subsidiary method in repairing the fire-damaged concrete.

Reviewer 2 Report
The paper seeks to introduce an approach ‘’ Recovery behavior of the macro cracks in elevated-temperature-damaged concrete after post-fire-curing”. However, the authors should consider improving upon the quality to further highlight and emphasize.
1. Based on the understanding of what entails in an abstract, consider adding one or two lines highlighting the significance of the study.
2. Tabulate all the materials used in this study with their respective physical and chemical properties.
3. Put space between any variable and its respective unit. For instance, the abstract contains “20°C” and “95%” instead of 20 °C and 95 % in line 134, respectively. consider correcting such anomalies throughout the article.
4. The introduction needs to be improved by relating to the mechanics of the studied materials and their mechanical characteristics. The references to be included: 10.1177/0731684417727143, 10.3390/polym14132662 and 10.1016/j.jiec.2022.06.023.
5. The statement “It is believed that the unequal coefficient of thermal expansion between the aggregate and the cement matrix.” in line 139 is incomplete and therefore does not make sense. Revise the sentence and come out with a meaningful statement relevant to the work.
6. What is the name of the device designed to determine the thermal expansion of the concrete and the mortar? Did you follow any standards in devising the device?
7. the font size of the figure inside Figure 5 is not visible or blurred. Edit the image and increase the font size including the numbering and then reinsert.
8. Add some figures from your results to the abstract.

Author Response
We are grateful for your useful advice and comments. We have revised the paper following your advice and comments, which we believe largely improved the quality of our paper. since the repsonse were a little long, please see the detailed response in the attached file.

Reviewer 3 Report
The article addresses very important issues of concrete durability after exposure to high temperature. The authors analyze microcracks in concrete subjected to annealing up to 800 degrees. C followed by the curing process. Below are the main notes for the work that require clarification and correction of the text: 1. Why was the method of hardening concrete samples with water only after they had cooled down? Please refer to the situation of extinguishing fires of a construction that is still on fire. 2. Please provide the graining curve of the aggregate used to make the concrete. 3. Please explain why thermal expansion tests were not performed on concrete only on mortars. This is how it is described in point 2.3. Then suddenly the article shows seed drills for concrete samples. 4. The text is carefully written. Comments are made when quoting publications in the text that are not adequate to the content. For example, the text line 239, 240 refers to publication [20] and in Fig. 6b there is reference to publication [21]. Please verify the quoted publications 5. In the description of the structure of cracks instead of cracks in the area of ​​the mortar, it is correct to use cracks in the cement matrix. 6. The photos in Fig. 4a are completely illegible. Please correct the drawing by showing a sample image for one sample on a larger scale, to reflect the essence of the measurement and analysis. No quantitative analysis of the observed figs was performed. 7. Better results can be obtained with computed microtomography. The authors should refer to this method in the introduction. The analysis carried out by the authors focuses only on macro-figures, which makes the issue much simpler. The paper requires more substantive correction and text improvement. It cannot be printed in Materials as it is.
Author Response
Thank you very much for the efforts you have made and the very useful suggetions and comments to our paper. we have revised the manuscript following your advice. However, for some comments, we truly respect and are grateful, we have given our explainsions and hope to dicuss. Please find the detailed repsonse in the attached file.

Reviewer 4 Report
Comments and Suggestions for Authors
The paper reports an interesting and very useful experimental work covering behavior of the macro cracks in elevated-temperature-damaged concrete. The manuscript is well structured and can be published after some revisions. The reviewer enjoyed reading this paper.
The manuscript has some weaknesses. Mentioned below aspects should be taken into consideration during the revision:
11. Units and abbreviations:
I suggest adding "Nomenclature" (as list of symbols, list of abbreviations and subscripts and others) in the manuscript.
22. Intoduction:
Literature analysis should be expanded. It is recommended to better justify fracture mechanisms investigations and fracture surface topography analysis. See for example the fallowing papers (Becker and Lampman 2002; Macek 2022):
- Becker WT, Lampman S (2002) Fracture Appearance and Mechanisms of Deformation and Fracture. In: ASM Handbook, Volume 11: Failure Analysis and Prevention
- Macek W (2022) The impact of surface slope and calculation resolution on the fractal dimension for fractures of steels after bending-torsion fatigue. Surface Topography: Metrology and Properties 10:015030. https://doi.org/10.1088/2051-672X/AC58AE
33. Experimental program:
a. Do the authors have SEM fractographies for the tested specimens? This could help to better understand the failure mechanisms.
b. Morphology and metrology of the fracture surface - is it possible to measure the surface topography for analysis? Do you have information about the surface roughness of fractures?
44. Conclusions:
a. The practical usefulness of the results should be emphasized.
b. The main limitations of the present method must be identified and discussed in the end of this section.

Author Response
Dear reviewer,
We are grateful for the efforts you have made to review our paper and thanks for your very professional comments. We have revised the manuscirpt following your advice and we believe that the paper is largely improved in quality. Please see the detailed response in the attached file.

Round 2
Reviewer 3 Report
Most of the authors made the necessary corrections. Article may be published in Materials.